# RadarOcc: Robust 3D Occupancy Prediction with 4D Imaging Radar

**Fangqiang Ding**[1,*]   **Xiangyu Wen**[1,*]   **Yunzhou Zhu**[2]   **Yiming Li**[3]   **Chris Xiaoxuan Lu**[4,†]

[1]University of Edinburgh    [2]Georgia Institute of Technology    [3]New York University
[4]AI Centre, Department of Computer Science, UCL

## Abstract

3D occupancy-based perception pipeline has significantly advanced autonomous driving by capturing detailed scene descriptions and demonstrating strong generalizability across various object categories and shapes. Current methods predominantly rely on LiDAR or camera inputs for 3D occupancy prediction. These methods are susceptible to adverse weather conditions, limiting the all-weather deployment of self-driving cars. To improve perception robustness, we leverage the recent advances in automotive radars and introduce a novel approach that utilizes 4D imaging radar sensors for 3D occupancy prediction. Our method, RadarOcc, circumvents the limitations of sparse radar point clouds by directly processing the 4D radar tensor, thus preserving essential scene details. RadarOcc innovatively addresses the challenges associated with the voluminous and noisy 4D radar data by employing Doppler bins descriptors, sidelobe-aware spatial sparsification, and range-wise self-attention mechanisms. To minimize the interpolation errors associated with direct coordinate transformations, we also devise a spherical-based feature encoding followed by spherical-to-Cartesian feature aggregation. We benchmark various baseline methods based on distinct modalities on the public K-Radar dataset. The results demonstrate RadarOcc's state-of-the-art performance in radar-based 3D occupancy prediction and promising results even when compared with LiDAR- or camera-based methods. Additionally, we present qualitative evidence of the superior performance of 4D radar in adverse weather conditions and explore the impact of key pipeline components through ablation studies.

## 1   Introduction

The safety of autonomous vehicles navigating in the wild hinges on a thorough understanding of the environment's 3D structure. As a unified scene representation built from grid-based volumetric elements known as voxels, 3D occupancy has gained increasing attention within the autonomous driving community [1–5]. Its rising popularity stems from its comprehensive scene depiction, capturing both geometric and semantic aspects. Crucially, it transcends the limitations of foreground-only representations (*vs.* 3D object detection [6–8]) and sparse data formats (*vs.* point cloud segmentation [9–11]). Furthermore, 3D occupancy offers a detailed open-set depiction of scene geometry, effectively handling out-of-vocabulary items (e.g., animals) and irregular shapes (e.g., cranes). This capability allows it to address a broader range of corner cases than previous object-based perception approaches [12–14].

Previous research has predominantly utilized either LiDAR point clouds [2, 15–22], RGB images [4, 5, 23–33], or a combination of both [3] for 3D occupancy prediction. However, the potential of 4D

---

[*]Equal contribution
[†]Corresponding author. Email: xiaoxuan.lu@ucl.ac.uk

38th Conference on Neural Information Processing Systems (NeurIPS 2024).

imaging radar [34, 35] —a critical sensor in autonomous driving—has been largely untapped in this area. Evolving from traditional 3D mmWave radars, this emerging sensor type enhances elevation resolution, enabling detection and resolution of targets across both horizontal and vertical planes, which results in detailed *imaging* outputs. Meanwhile, 4D radar inherits the traditional advantages of mmWave radar, such as capability in all lighting and weather conditions, object velocity measurement, and cost-effectiveness compared to LiDAR systems. These attributes, particularly its resilience in adverse weather conditions like fog and rain, position 4D radar as an essential component in achieving mobile autonomy.

In this work, we explore the potential of 4D imaging radar to enhance 3D occupancy prediction. Previous research in radar perception has largely relied on 4D radar point clouds as input, a method inspired by LiDAR techniques. This 'LiDAR-inspired' framework has demonstrated effectiveness in tasks such as 3D object detection and tracking [36–58]. However, this approach primarily enhances the detection of foreground objects such as cars, pedestrians, and trucks. In contrast, 3D occupancy prediction requires the detection of signal reflections from all occupied spaces, encompassing both foreground and background elements like roads, barriers, and buildings. The traditional reliance on sparse radar point clouds, therefore, is not optimal for 3D occupancy prediction, as critical environmental signals are often lost during the point cloud generation process [59, 60]. For instance, the surface of highways, typically made of low-reflectivity materials such as asphalt, often yields weak signals back to the radar receiver.

To avoid the loss of negligible signal returns, we propose utilizing the 4D radar tensor (4DRT) for 3D occupancy prediction. This raw data format preserves the entirety of radar measurements, offering a comprehensive dataset for analysis. However, employing such volumetric data introduces significant challenges. For instance, the substantial size of 4DRTs—potentially up to 500MB—poses processing inefficiencies that could compromise real-time neural network performance. Additionally, raw radar data is inherently noisy due to the multi-path effect and is stored in spherical coordinates, which diverges from the preferred 3D Cartesian occupancy grid used in our applications.

Motivated by the outlined challenges, we introduce a novel approach, `RadarOcc`, specifically tailored for 4DRT-based 3D occupancy prediction. To address the computational and memory demands, our method initially reduces the data volume of 4DRTs through the encoding of Doppler bins descriptors and implementing spatial sparsification in the preprocessing stages. Our technique features sidelobe-aware spatial sparsification to minimize the interference scattered across azimuth and elevation axes, which is further refined through range-wise self-attention mechanisms. Importantly, we observed the typical conversion of spherical RTs to Cartesian data volumes, which often incurs non-negligible interpolation errors. Instead, we directly encode spatial features in spherical coordinates and seamlessly aggregate them using learnable voxel queries defined in Cartesian coordinates. Our approach further employs 3D sparse convolutions and deformable attention [61] for efficient feature encoding and aggregation. `RadarOcc` is benchmarked on the K-Radar dataset [42] against state-of-the-art methods across various modalities, demonstrating the promising performance in radar-based 3D occupancy prediction. Comprehensive experiment results validate its comparable performance to the camera and LiDAR solutions. A qualitative assessment further validates the superior robustness of 4D radar data under adverse weather conditions, establishing its capability for all-weather 3D occupancy prediction. The contributions of this work are three-fold:

- Introduction of the first-of-it-kind method, `RadarOcc`, for 4D radar-based 3D occupancy prediction in autonomous driving. We recognize the limitation of radar point clouds in reserving critical raw signals and advocate the usage of 4DRT for occupancy perception.
- Development of a novel pipeline with techniques to cope with challenges accompanying 4DRTs, including reducing large data volume, mitigating sidelobes measurements and interpolation-free feature encoding and aggregation.
- Extensive experiments on the K-Radar dataset, benchmarking state-of-the-art methods based on different modalities, and validating the competitive performance of `RadarOcc` and its robustness against adverse weather. We release our code and model at `https://github.com/Toytiny/RadarOcc`.

## 2 Related work

**3D occupancy prediction.** Early attempts on 3D occupancy prediction, *aka.* semantic scene completion (SSC) [62], are mainly limited to the small-scale interior scenes [62–71]. The introduction

of SemanticKITTI [72] expands the study of SSC to large-scale outdoor scenes, based on which some works validate the feasibility of outdoor SSC with LiDAR input [15–19]. In contrast, MonoScene [23] is the seminal work for SCC using only a single monocular RGB image. Since Tesla's disclosure of their occupancy network for Full Self-Driving (FSD) [1], there has been a recent surge of research on 3D occupancy prediction for autonomous vehicles. While a few works leverage LiDAR point clouds [2, 3, 20–22] for scene completion, the majority of existing approaches rely on a vision-only pipeline that learns to lift 2D features into the 3D space [3–5, 24–33]. Despite these prevalent solutions based on LiDAR and camera, 4D radar sensors are still under-explored for 3D occupancy prediction.

**4D radar for autonomous driving.** As an emerging automotive sensor, 4D mmWave radar prevails over LiDAR and camera in adverse weather (*e.g.*, fog, rain and snow), offering all-weather sensing capabilities for mobile autonomy. In recent years, increasing endeavours have been witnessed to unveil the potential of 4D radar for autonomous driving applications, encompassing 3D object detection [36–57] and tracking [56–58], scene flow estimation [73–75], odometry [73, 76–82] and mapping [80–82]. Apart from these works, we are the pioneering study for 4D radar-based 3D occupancy prediction, further exploring this unique sensor for the untouched topic.

**Radar tensor for perception**  Besides the post-processing radar point cloud, another data type of mmWave radar is the radar tensor (RT), which is the product of applying FFT along the corresponding dimensions to the raw ADC samples (*c.f.* Sec. 3.1). Unlike the sparse radar point cloud, dense RTs contain rich and complete measurements of the environment, refraining from information loss during point cloud generation (*e.g.*, CFAR [59, 60]). Consequently, some works attempt to use 2D [37, 83–86], 3D [87–89] or 4D [42, 46, 52] RTs for object detection, yielding satisfactory performance. In this work, we develop a tailored approach to 4D radar-based 3D occupancy prediction based on 4DRTs.

# 3   Preliminary

## 3.1   4D radar signal processing pipeline

**ADC samples.** To measure the surroundings, a sequence of FMCW waveforms, aka. chirp signals, are emitted by the transmit (Tx) antennas within a short timeframe. These signals are reflected off objects and captured by the receive (Rx) antennas. The intermediate frequency (IF) signal is produced by mixing the signals from a Tx-Rx antenna pair. This mixed signal is then sampled by an Analog-to-Digital Converter (ADC) to generate discrete samples for each chirp [90]. By compiling ADC samples from all chirps and Tx-Rx antenna pairs, the FMCW radar system constructs a 3D complex data cube for each frame. This data cube is organized into three dimensions: *fast time*, *slow time*, and *channel*, which correspond to range, range rate, and angle, respectively [91].

**Radar tensor.** Utilizing ADC samples, Fast Fourier Transforms (FFTs) are applied across relevant dimensions to extract detailed information. The first FFT, known as range-FFT, is performed across the sample (fast time) axis to separate objects at different distances into distinct frequency responses within range bins defined by hardware specifications. Subsequently, a Doppler-FFT along the chirp (slow time) axis decodes phase variances—Doppler bins—to derive relative radial velocities, producing a range-Doppler heatmap. For configurations with multiple Rx-Tx antenna pairs, termed *virtual* antenna elements, additional FFTs (angle-FFT) are executed across the spatial dimensions of the virtual antenna array to determine Angles of Arrival (AoA) for azimuth and elevation angles. This series of transformations results in a comprehensive 4D radar tensor (4DRT), characterized by power measurements across range, Doppler velocity, azimuth, and elevation dimensions.

**Radar point cloud.** Beyond analyzing radar tensors, most FMCW radar sensors further refine their output to identify salient targets, which typically represent less than 1% of the data. Target detection algorithms such as CA-CFAR [59] and OS-CFAR [92] are commonly applied to the range-Doppler heatmap [91, 93] or directly on the 3D/4D radar tensors [42, 46] to isolate peak measurements. This process generates a sparse radar point cloud, with each point characterized by 3D coordinates and attributes such as Doppler velocity, power intensity, or radar cross-section (RCS). While this step significantly reduces data volume and mitigates noise, it also eliminates a substantial amount of potentially valuable information.

### 3.2 4DRT for 3D occupancy prediction

**Rationale of using 4DRT.** 4D radar tensors (4DRTs) serve as raw sensor data that amalgamate the strengths of LiDAR/radar point clouds and RGB images, providing direct 3D measurements in a continuous data format. These tensors comprehensively capture information from raw radar measurements, effectively addressing the shortcomings associated with the sparseness of radar point clouds caused by the signal post-processing. For instance, low-reflectivity surfaces like asphalt, common on highways, typically do not reflect enough radar signals for detection. By using 4DRTs, these minimal signal returns can be detected, significantly bolstering occupancy prediction capabilities. Furthermore, the volumetric structure of 4DRTs aligns well with 3D occupancy grids, making them ideally suited for advancing 3D occupancy prediction techniques.

**Challenges.** Despite their significant advantages, using 4D radar tensors (4DRTs) for 3D occupancy prediction presents substantial challenges. First, the large data size of 4DRTs (e.g., 500MB per frame in the K-Radar dataset [42]) hinders computational efficiency, necessitating data volume reduction before processing. Second, the inherent noise in radar data, exacerbated by the multi-path effect of mmWave, requires careful filtering to preserve essential signals while eliminating noise. Third, the discrepancy between the spherical coordinates of 4DRT data and the Cartesian coordinates required for 3D occupancy outputs calls for a tailored network design. This design must effectively translate spatial interactions from spherical to Cartesian dimensions to ensure accurate occupancy predictions.

## 4 Method

### 4.1 Task definition

In this work, we consider the task of 3D occupancy prediction with single-frame 4DRT output from 4D imaging radar. Given a 4DRT captured in the current frame denoted as $\mathbf{V} \in \mathbb{R}^{R \times A \times E \times D}$, our task aims to predict a 3D volume $\mathbf{O} = \{o_i\}_{i=1}^{H \times W \times L}$, of which each voxel element $o_i \in \{c_0, c_1, \ldots, c_C\}$ is represented as either free (*i.e.,* $c_0$) or occupied with a certain semantics $c_i(i > 0)$ out of $C$ classes. Here, $R$, $A$, $E$, and $D$ denote the number of bins along the range, azimuth, elevation and Doppler axis, respectively, and each scalar of the 4DRT is the power measurement mapped to a location within the space defined by these four axes. $H$, $W$ and $L$ represent the volumetric size of the predefined region of interest (RoI) in the height, width and length dimensions.

### 4.2 Overview

`RadarOcc` consists of four components in tandem (*c.f.* Fig. 1). Before loading heavy 4DRTs to the neural network, we reduce their data volume as the preprocessing steps via encoding the Doppler bins descriptor and performing sidelobe-aware spatial sparsifying to improve the efficiency without losing the key information (*c.f.* Sec. 4.3). To refrain from the interpolation error, we encode spatial features directly on the spherical RTs without transforming them into Cartesian volumes (*c.f.* Sec. 4.4) and aggregate the spherical features with 3D volume queries defined in the Cartesian coordinates (*c.f.* Sec. 4.5). Specifically, range-wise self-attention is used to alleviate the sidelobes, and sparse convolution and deformable attention are leveraged for fast feature encoding and aggregation. The occupancy probabilities are predicted in the 3D occupancy decoding step, which is supervised via our training loss (*c.f.* Sec. 4.6).

### 4.3 Data volume reduction

Direct processing of raw 4DRTs with neural networks is impractical due to its substantial data size (*e.g.*, 500MB per frame) which leads to heavy computation cost and memory usage. Moreover, the slow data transfer between the sensor, storage device and processing unit (CPU/GPU) of large-volume raw 4DRTs not only hinders the onboard runtime efficiency but also increases the training duration which demands repetitive data loading. For efficiency, we propose to reduce the data volume of 4DRTs through encoding the Doppler bins descriptor and sidelobe-aware spatial sparsifying as the preprocessing steps (see Fig. 1). Post reduction, the loading of 4DRTs into the processing unit for runtime inference can be more feasible and the network training can be more efficient.

**Doppler bins descriptor.** Unlike the three spatial axes, which are intuitively critical for spatial perception, the Doppler axis in 4DRTs has often been considered redundant in 3D object detection.

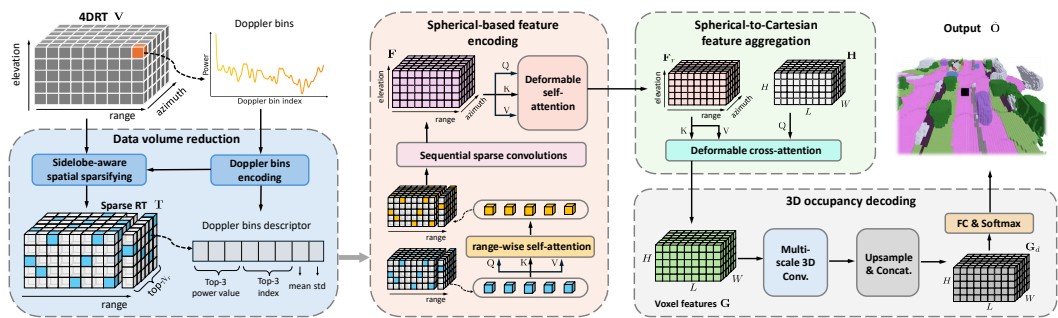

Figure 1: Overall pipeline of RadarOcc. The data volume reduction pre-processes the 4DRT into a lightweight sparse RT via Doppler bins encoding and sidelobe-aware spatial sparifying. We apply spherical-based feature encoding on the sparse RT and aggregate the spherical features using Cartesian voxel queries. The 3D occupancy volume is finally output via 3D occupancy decoding.

Previous studies [42, 46, 52] have employed average-pooling to minimize this axis, aiming to reduce computational overhead. However, we argue that this ostensibly 'redundant' axis contains vital cues for geometric and semantic analysis in 3D occupancy prediction. Specifically, the Doppler axis provides essential information on object speed via peak locations, aiding in differentiating dynamic objects from static backgrounds. Moreover, the power distribution within the Doppler bins offers insights into the confidence levels of true targets—essentially, indicating their likelihood of occupancy. To preserve and utilize this crucial information, we have developed a method to encode the Doppler bins into a descriptor that captures specific statistics for each spatial location within the 4DRTs. This descriptor incorporates the top-three power values along with their indices, the mean power value, and the standard deviation, as depicted in Fig 1. Note that the number of preserved top values is determined empirically. Consequently, this approach enables us to reduce the data volume of raw 4DRTs by a factor of $\frac{D}{8}$, while retaining key information from the Doppler axis.

**Sidelobe-aware spatial sparsifying.** By encoding the Doppler bins into light-weight descriptors, we transform the raw 4DRT into 3D spatial data volume with the original Doppler axis as the 8-channel feature dimension. Nevertheless, it remains costly for neural networks to encode features from 3D dense data volume with operations like 3D convolution [94, 95]. To accelerate the computation, prior arts [42, 46] transfer the dense RT into a sparse format by retraining only the top-percentile elements based on power measurements. However, this approach tends to be biased towards specific ranges that exhibit exceptionally high measurements. It can be observed in Fig. 2 that after percentile-based sparsifying, a significant number of the reserved elements are concentrated within the same ranges spread across the azimuth and elevation axes. These elements manifest as artifacts of sidelobes, which can an be viewed as the diffraction pattern of the antenna [96, 97]. Consequently, this results in the loss of important measurements from other ranges and introduces lots of noise into the sparse tensor. To mitigate this issue, we propose to select the top-$N_r$ elements for each individual range instead of on the whole dense RT for spatial sparsifying (see Fig. 1). In this way, the dominance of certain ranges can be avoided while the sidelobe level is reduced, as exhibited in Fig. 2. Note that our spatial element selection is based on the mean power value across the Doppler axis. The final sparse tensor is denoted as $\mathbf{T} = \{t_i \in \mathbb{R}^{N_r \times (8+2)}\}_{i=1}^{R}$ with the extra two feature channels storing the azimuth and elevation indices of reserved $N_r$ elements for each range.

### 4.4 Spherical-based feature encoding

Given the sparse RT, we aim to encode representative features for accurate 3D occupancy prediction. As the sparse RTs are inherently in the spherical coordinates, previous works [42, 46] transfer them into the Cartesian coordinates before feature encoding. However, such a transfer would undermine their uniform density distribution and often incur interpolation errors. Inspired by the polar representation of point clouds [10, 98, 99], we propose to take the elements in RT as voxels rasterized in the spherical coordinates and apply the spherical-based feature encoding directly. The spherical voxel representation naturally matches the spherical-uniform distribution of RTs and can refrain from inducing interpolation errors. In practice, the 3D convolutions can be used to extract grid-based representations by only replacing the $X$-$Y$-$Z$ axis with the range-azimuth-elevation axis. In what follows, we illustrate our spherical-based feature encoding process.

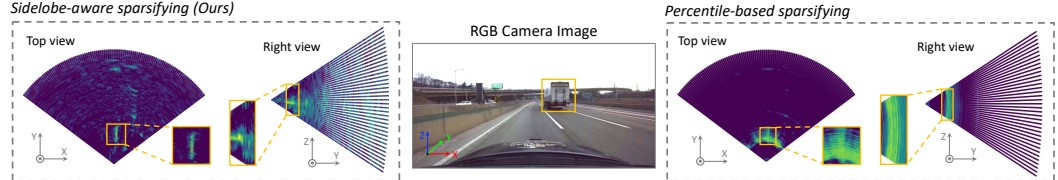

Figure 2: Comparison between the sparse RTs resulted by our sidelobe-aware and percentile-based sparsifying [42, 46]. We transform the spherical RT elements to the Cartesian coordinates and show them in two views. The arches on the heatmap indicate the same ranges. Percentile-based method retains many elements caused by sidelobe noise, which are concentrated at certain ranges. In contrast, our method can reduce the sidelobe level and reserve critical measurement from different ranges.

**Range-wise self-attention.** In Section 4.3, we address the issue of sidelobes by selecting elements based on range-wise percentiles during the preprocessing phase. To further mitigate sidelobe interference, we introduce a range-wise self-attention mechanism [100] (see Fig. 1) as the initial step in our feature encoding process. Specifically, within each range component $t_i \in \mathbf{T}$, which includes $N_r$ RT tokens, we utilize the Doppler bin descriptors as token features. Additionally, two index channels are employed for positional embeddings to enhance the specificity of our spatial encoding.

**Sequential sparse convolution.** For efficiency, we apply a series of 3D sparse convolutions [101] onto the sparse RT for spatial feature encoding in the spherical voxel space. This produces a 3D dense feature volume $\mathbf{F} \in \mathbb{R}^{\frac{R}{S} \times \frac{A}{S} \times \frac{E}{S} \times C_f}(N_f = \frac{R}{S} \times \frac{A}{S} \times \frac{E}{S})$ with a reduce spatial dimension characterized by the stride $S$, where $C_f$ denotes the feature dimension. Note that $\mathbf{F}$ inherently aligns with the spherical space with each feature element's indices corresponding to a spherical coordinate.

**Deformable self-attention.** Following the consecutive 3D sparse convolution, we use the 3D deformable attention [61] to further refine and augment our feature volume $\mathbf{F}$ by enforcing spatial interaction. As a definition, for a query feature $z$ corresponding to a reference point $p$ in the input feature $\mathbf{X}$, its feature can be updated by deformable attention in the following equation:

$$\text{DeformAttn}(z, p, \mathbf{X}) = \sum_{m=1}^{M} \mathbf{W}_m \left[ \sum_{k=1}^{K} \mathbf{A}_{mk} \cdot \mathbf{W}'_m \mathbf{X}(p + \Delta p_{mk}) \right] \quad (1)$$

where $\mathbf{W}_m$ and $\mathbf{W}'_m$ are the learnable weights for the $m$-th attention head, while $\mathbf{A}_{mk}$ and $\Delta p_{mk}$ is the attention weight and sampling offset calculated with $z$ for its $k$-th sampling point and the $m$-th head. $\mathbf{X}(p + \Delta p_{mk})$ is the key features at the sample location $(p + \Delta p_{mk})$. By applying self-attention to $\mathbf{F} = \{f^q\}_{q=1}^{N_f}$, the refined feature volume $\mathbf{F}_r = \{f_r^q\}_{q=1}^{N_f}$ can be derived by:

$$f_r^q = \text{DeformAttn}(f^q, p^q, \mathbf{F}) \quad (2)$$

### 4.5 Spherical-to-Cartesian feature aggregation

Decoding 3D Cartesian occupancy from a spherical feature volume is inherently challenging due to misalignments in spatial axes and discrepancies in the regions they represent. An intuitive approach would be to transform the spherical feature volume into a Cartesian one and then decode the 3D Cartesian occupancy. However, this method can introduce feature-level interpolation errors, which we aim to avoid as discussed in Section 4.4.

To avoid conducting interpolation, we propose to aggregate the spherical features in a *learnable* way, with 3D volume queries defined in the Cartesian coordinates attending to the feature samples in $\mathbf{F}_r$, as shown in Fig. 1. First, we build learnable grid-based voxel queries $\mathbf{H} = \{h^q \in \mathbb{R}^{C_f}\}_{q=1}^{H \times W \times L}$ which has the same volumetric size as our desired output $\mathbf{O}$ and the same feature dimension as the spherical feature volume $\mathbf{F}_r$. Each voxel query $h^q$ corresponds to a 3D point $p^q$ in the Cartesian coordinate. Second, the 3D point $p^q$ of each query is transformed from the Cartesian to the spherical coordinate, which is then mapped to a index position in $\mathbf{F}_r$ denoted as $\Phi(p^q)$. We take $\Phi(p^q)$ as a 3D reference point in the spherical space and sample key elements in its vicinity from the feature volume $\mathbf{F}_r$. Lastly, we leverage deformable cross-attention [61] to aggregate the key samples for each reference point and the output $\mathbf{G} = \{g^q \in \mathbb{R}^{C_f}\}_{q=1}^{H \times W \times L}$ can be calculated by:

$$g^q = \text{DeformAttn}(h^q, \Phi(p^q), \mathbf{F}_r) \quad (3)$$

### 4.6   3D occupancy decoding and supervision

With the aggregated voxel features $\mathbf{G}$, we leverage consecutive 3D convolutions [94, 95] with skip connection [102] to decode hierarchical feature volumes at $N_s$ scales with a scaling step of 2. Multi-scale feature volumes are then merged in a top-down way [103] via upsampling features by a factor 2 and concatenated along the feature dimension, resulting in $\mathbf{G}_d \in \mathbb{R}^{H \times W \times L \times N_s C_f}$. Finally, the occupancy head equipped with the *softmax* function is employed to output the normalized occupancy probabilities $\tilde{\mathbf{O}} \in \{0,1\}^{H \times W \times L \times (C+1)}$ for all voxels on $C$ semantic classes and one free class.

Our network is trained in a supervised way with the ground truth occupancy. Following [3], we use the cross-entropy loss as the primary loss to optimize the training and incorporate the lovasz-softmax loss [104] to handle the class imbalances. Moreover, we utilize the scene- and class-wise affinity loss proposed in [23] to enhance the optimization of geometry and semantic IoU metrics.

## 5   Experiment

### 5.1   Experimental setup

**Dataset preparation.**  Our experiments are conducted on the K-Radar dataset [42], which is, to the best of our knowledge, the only autonomous driving dataset providing available 4DRT data. Besides, K-Radar also contains multi-modal data from LiDAR, camera, GPS-RTK and annotated 3D bounding boxes and tracking IDs, enabling us to compare between different modalities and generate 3D occupancy labels. Following [3, 25, 105], we generate occupancy ground truth by superimposing consecutive LiDAR sweeps and construct the dense 3D occupancy grids via voxelization. To handle scene dynamics, we register objects with the same tracking IDs across the sequence. As K-Radar does not annotate fine-grained point-level semantics, we segment the scene into the foreground (*e.g.,* sedan, truck, pedestrian) and background using bounding boxes and label the voxel grids into three classes, including foreground, background and free. Many sequences in K-Radar were collected under adverse weather (*i.e.,* sleet, rain, and snow), which results in non-negligible noise to the generated occupancy labels based on LiDAR sweeps. Therefore, we reserve this adverse-weather test split for qualitative comparison and only generate the occupancy labels for the well-condition sequences, which are separated into the training, validation and test splits.

**Evaluation protocol.**  As the pioneering study of 3D occupancy prediction using the K-Radar dataset, we have tailored the evaluation protocol to align with our experimental needs. We define the Region of Interest (RoI) with specific dimensions: a front range of [0, 51.2m], a side range of [-25.6m, 25.6m], and a height range of [-2.6m, 3m]. The voxel resolution is set at 0.4m, resulting in a target occupancy volume of $128 \times 128 \times 14$ voxels. Consistent with established methods in the field [3, 72, 105], we employ the Intersection over Union (IoU) metric to evaluate the geometric accuracy of our occupancy predictions, focusing solely on the occupied or free status without integrating semantics. Additionally, to gauge the effectiveness of our foreground-background segmentation, we calculate the mean IoU (mIoU) across these two classes. In line with previous studies [5, 105], we present our findings across multiple ranges, specifically at 51.2m, 25.6m, and 12.8m.

**Competing methods.**  We benchmark `RadarOcc` against state-of-the-art methods employing different modalities. Given that recent studies do not use radar data for 3D occupancy prediction, we adapt the OpenOccupancy LiDAR-based baseline and CONet [3] to accommodate radar point cloud (RPC) inputs for our comparative analysis. Furthermore, we convert 4DRTs to Cartesian coordinates [42] with a voxel size of 0.4m, referred to as 4DRT-XYZ, and integrate them into the LiDAR-based OpenOccupancy framework [3]. Following best practices from [42, 46], we process 4DRT-XYZ into a sparser format. For a comprehensive inter-modality evaluation, we also replicate the OpenOccupancy LiDAR-based baseline [3] and both monocular and stereo camera-based SurroundOcc [25] configurations to fit our experimental setup. Notably, we enrich our comparisons by generating 16-beam and 32-beam LiDAR point clouds from the standard 64-beam configurations through elevation-wise downsampling. The evaluation focuses on the overlap area between the horizontal field of view (FoV) of all sensors and our defined RoI to minimize potential data discrepancies beyond the FoV. For implementation, we train all evaluated models on our K-Radar well-condition training set.

| | | IoU (%) | | | mIoU (%) | | | 🟩 BG IoU (%) | | | 🟥 FG IoU (%) | | |
|---|---|---|---|---|---|---|---|---|---|---|---|---|---|
| Method | Input | 12.8m | 25.6m | 51.2m | 12.8m | 25.6m | 51.2m | 12.8m | 25.6m | 51.2m | 12.8m | 25.6m | 51.2m |
| L-baseline [3] | RPC | 42.8 | 34.9 | 27.9 | 23.5 | 18.6 | 14.6 | 43.5 | 34.6 | 27.3 | 3.5 | 2.6 | 1.9 |
| L-CONet [3] | RPC | 46.1 | 36.0 | 25.0 | 24.6 | 20.3 | 14.4 | 43.3 | 35.4 | 25.6 | 5.8 | 5.2 | 3.1 |
| L-baseline [3] | 4DRT-XYZ | 47.4 | 38.1 | 28.5 | 29.9 | 24.3 | 17.5 | 46.4 | 37.5 | 27.9 | 13.4 | 11.1 | 7.2 |
| RadarOcc (Ours) | 4DRT | **48.8** | **39.1** | **30.4** | **34.3** | **28.5** | **22.6** | **47.9** | **38.2** | **29.4** | **20.7** | **18.7** | **15.8** |

Table 1: Quantitative comparison between `RadarOcc` and state-of-the-art radar-based baseline methods. Results are reported on K-Radar well-condition test split. Best result is shown in **bold**.

| | | IoU (%) | | | mIoU (%) | | | 🟩 BG IoU (%) | | | 🟥 FG IoU (%) | | |
|---|---|---|---|---|---|---|---|---|---|---|---|---|---|
| | Method | 12.8m | 25.6m | 51.2m | 12.8m | 25.6m | 51.2m | 12.8m | 25.6m | 51.2m | 12.8m | 25.6m | 51.2m |
| (a) | Ours | **48.8** | 39.1 | **30.4** | **34.3** | 28.5 | **22.6** | **47.9** | 38.2 | **29.4** | **20.7** | 18.7 | 15.8 |
| (b) | Ours w/o DBD | 48.1 | **39.4** | 30.0 | 33.6 | **28.9** | 22.6 | 47.2 | **38.7** | 29.2 | 20.0 | **19.1** | **16.0** |
| (c) | Ours w/o SSS | 44.2 | 36.8 | 28.7 | 24.1 | 20.2 | 15.6 | 42.3 | 35.6 | 27.6 | 5.9 | 4.7 | 3.5 |
| (d) | Ours w/o SFE | 46.2 | 38.4 | 29.4 | 30.4 | 26.5 | 21.1 | 45.5 | 37.5 | 28.5 | 15.4 | 15.5 | 13.9 |

Table 2: Ablation studies on key designs of `RadarOcc`. DBD, SSS, SFE refer to the Doppler bins descriptor, sidelobe-aware spatial sparfiying, and spherical-based feature encoding, respectively.

## 5.2 Comparison against radar-based methods

We first compare `RadarOcc` with state-of-the-art baseline methods using radar data for 3D occupancy prediction in Tab. 1. As can be seen, `RadarOcc` outperforms other approaches in every metric, demonstrating its state-of-the-art performance in radar-based 3D occupancy prediction. Specifically, our 4DRT-based `RadarOcc` largely improves the performance over RPC-based methods: the mIoU of L-CONet [3] is relatively improved by 39.4%, 40.4% and 56.9% for different volumes (12.8m, 25.6m, 51.2m). Such a significant improvement mainly stems from the dense data format of 4DRT, which retains critical information from low-reflectivity objects, enabling effective occupancy prediction for the whole scene. 4DRT-XYZ based L-baseline [3] also outperforms RPC-based methods but inferior to `RadarOcc`, especially in long-range FG IoU. We credit this to the interpolation errors led to small and far foreground objects when we converting 4DRT to Cartesian coordinates.

## 5.3 Ablation study

To validate the effectiveness of our key designs, we ablate them alone from our 4DRT-based pipeline `RadarOcc` and show the evaluation results on K-Radar well-condition test split in Tab. 2.

**Doppler bins descriptor.** By replacing the Doppler bins descriptor with the average-pooling result, the performance of `RadarOcc` is degraded in most metrics (row (a) vs. (b) in Tab. 2), demonstrating the usefulness of preserving the information encoded by the Doppler axis (*c.f.* Sec. 4.3). However, the improvement is somehow marginal due to the limited Doppler measurement range of the radar used in K-Radar [42], which wraps around the overflow values, causing ambiguity in Doppler velocity.

**Sidelobe-aware spatial sparsifying.** We conduct this experiment (row (c) in Tab. 2) by changing our sidelobe-aware spatial sparsifying (*c.f.* Sec. 4.3) to the percentile-based spatial sparsifying used in [42, 46]. Our sidelobe-aware approach leads to a remarkable advancement in performance, especially in mIoU metrics. This is attributed to its ability to preserve more valid elements from diverse ranges and suppress sidelobes for sparse RTs, allowing for more accurate prediciton.

**Spherical-based feature encoding.** For row (d) in Tab. 2, we transform sparse RT to Cartesian coordinates before feature encoding (*c.f.* Sec. 4.4) and omit the spherical-to-Cartesian feature aggregation (*c.f.* Sec. 4.5). We can see that our spherical-based feature encoding gains the performance for each metric as our strategy preserves the original data distribution, avoiding incurring interpolation errors. This also validates the effectiveness of our learnable spherical-to-Cartesian feature aggregation.

## 5.4 Model efficiency

To assess the runtime efficiency of `RadarOcc`, we conducted our model inference on a single Nvidia GTX 3090 GPU. The results shows an average inference speed of approximately 3.3fps. Although

| Method | range-wise attn. | seq. sparse conv. | deform. self-attn. | deform. cross-attn. | occ. decoding | total runtime | fps |
|---|---|---|---|---|---|---|---|
| RadarOcc | 2.5 | 47.5 | 88.8 | 72.0 | 92.1 | 302.9 | 3.30 |
| RadarOcc (w. optim.) | 2.5 | 20.7 (-56.4%) | 32.8 (-63.1%) | 29.7 (-58.7%) | 48.3 (-47.6%) | 133.9 (-55.8%) | 7.46 (+126.1%) |

Table 3: Comparison between `RadarOcc` and its lightweight version after computation optimization in terms of each component's and total runtime (ms) and fps. Relative change is shown in (·).

| Method | IoU (%) | | | mIoU (%) | | | 🟩 BG IoU (%) | | | 🟥 FG IoU (%) | | |
|---|---|---|---|---|---|---|---|---|---|---|---|---|
| | 12.8m | 25.6m | 51.2m | 12.8m | 25.6m | 51.2m | 12.8m | 25.6m | 51.2m | 12.8m | 25.6m | 51.2m |
| RadarOcc | **48.8** | **39.1** | **30.4** | 34.3 | **28.5** | **22.6** | **47.9** | **38.2** | **29.4** | 20.7 | **18.7** | **15.8** |
| RadarOcc (w. optim.) | 46.5 | 38.0 | 29.3 | **35.5** | 27.6 | 20.9 | 46.0 | 37.6 | 28.8 | **25.0** | 17.5 | 13.1 |

Table 4: Comparison between `RadarOcc` and its lightweight version after computation optimization in terms of performance across metrics at different ranges. Better result is shown in **bold**.

| Method | Input | IoU (%) | | | mIoU (%) | | | 🟩 BG IoU (%) | | | 🟥 FG IoU (%) | | |
|---|---|---|---|---|---|---|---|---|---|---|---|---|---|
| | | 12.8m | 25.6m | 51.2m | 12.8m | 25.6m | 51.2m | 12.8m | 25.6m | 51.2m | 12.8m | 25.6m | 51.2m |
| L-baseline [3] | L (16) | 49.1 | 43.3 | 35.2 | 39.0 | 34.3 | 28.2 | 48.2 | 42.5 | 34.4 | 29.8 | 26.1 | 22.1 |
| | L (32) | 51.1 | 44.0 | 34.9 | 42.1 | 35.0 | 28.9 | 50.8 | 43.6 | 34.2 | 33.5 | 26.3 | 23.6 |
| | L (64) | 56.9 | 52.5 | 43.8 | 53.7 | 45.2 | 36.6 | 56.1 | 51.8 | 43.3 | 51.2 | 36.5 | 29.9 |
| SurroundOcc [25] | C | 44.3 | 33.1 | 24.1 | 36.1 | 23.9 | 14.7 | 44.1 | 32.9 | 23.7 | 28.2 | 15.0 | 5.7 |
| | C (S) | 46.2 | 34.4 | 25.4 | 40.8 | 25.4 | 16.2 | 45.5 | 34.1 | 25.1 | 36.1 | 16.7 | 7.3 |
| RadarOcc (Ours) | 4DRT | 48.8 | 39.1 | 30.4 | 34.3 | 28.5 | 22.6 | 47.9 | 38.2 | 29.4 | 20.7 | 18.7 | 15.8 |

Table 5: Quantitative comparison between `RadarOcc` and state-of-the-art methods based on LiDAR and camera. Results are reported on K-Radar well-condition test split. (·) is the number of LiDAR beams and (S) denotes stereo. The top four methods are colored as **red**, **green**, **blue**, and **orange**.

there is still a gap between the real-time application (*i.e.,* 10fps), our inference speed has surpassed that of many camera-based methods as reported in [25]. Further improvements in inference speed can be achieved by reducing network complexity and applying precision reduction techniques, such as converting model precision from *Float32* (FP32) to *Float16* (FP16).

To validate this, we simplified the feature encoding (*c.f.* Sec. 4.4) and aggregation (*c.f.* Sec. 4.5) modules by reducing some redundancy layer (*e.g.*, number of layers in deformable attention) for efficiency, and converted the computationally intensive 3D occupancy decoding module (*c.f.* Sec. 4.6) from FP32 to FP16 via the quantization in PyTorch. These optimizations resulted in a 126% increase in inference speed, reaching approximately 7.46 fps, with only a minimal impact on performance. Please refer to Tab. 3 and Tab. 4 for detailed changes in runtime for each module and performance. Given the increasing computational power of modern embedded GPUs, such as the Nvidia Jetson Orin, which can almost rival desktop GPUs like the Nvidia GTX 2090, we believe this enhanced inference speed demonstrates the potential for real-time application of our method in future vehicle systems, especially if further model quantization is applied.

### 5.5 Comparison between different modalities

To enrich our benchmark results and provide insights into the performance comparison between different modalities, we also evaluate state-of-the-art baseline methods [3, 25] on LiDAR and camera input. Quantitative results on K-Radar well-condition test split are reported in Tab. 5, while examples of qualitative results on K-Radar adverse-weather testing splits are exhibited in Fig. 3.

**Quantitative results under normal weathers.** As seen in Tab. 5, not surprisingly, LiDAR-based L-baselines [3] rank the top three in most metrics thanks to LiDAR's low-noise and high-resolution measurements (*vs.* radar) and direct depth measurement (*vs.* camera). Due to the inherently lower resolution and considerable noise of radar data, radar-based methods exhibit inferior to LiDAR-based methods in normal weather. However, `RadarOcc` still shows comparable performance to 16-beam LiDAR, and surpasses monocular and stereo camera-based method in most metrics. Notably, `RadarOcc` outperforms state-of-the-art SurroundOcc [25] relatively by 39.5%/19.7% and 53.7%/26.1% in mIoU/IoU@51.2m for stereo and monocular input, respectively. Stereo camera-based SurroundOcc [25] ranks third on FG IoU and mIoU@12.8m because of stereo vision's ability to infer accurate depth at short ranges, where the disparity between the two images is more pronounced.

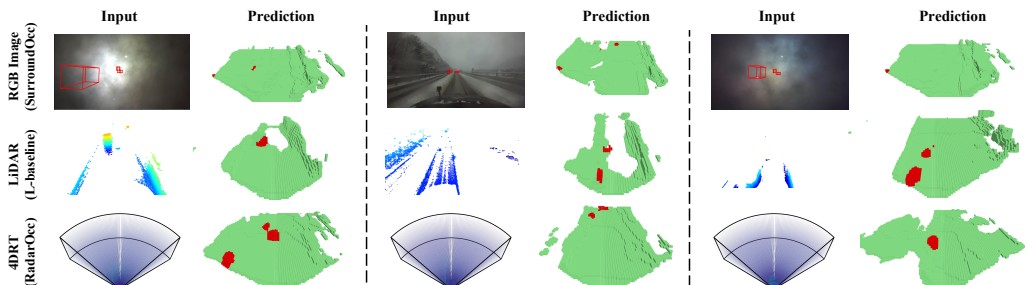

Figure 3: Qualitative comparison between `RadarOcc`, LiDAR-based L-baseline [3] and camera-based SurroundOcc [25] in adverse weathers. Ground truth bounding boxes are shown in RGB images.

**Qualitative results under adverse weathers.** While we have demonstrated the competitive performance of `RadarOcc` under normal weather, the key reason behind using radar for perception comes from its unique robustness against adverse weather where LiDAR and cameras fall short. To showcase such an inherent advantage, we provide some examples of qualitative results from different modalities in Fig. 3. As can be seen, `RadarOcc` provide robust 3D occupancy prediction under heavy rain and snow. In contrast, the camera lens are covered by the rain/snow and LiDAR measurements of some objects ahead are missing as water droplets or snowflakes can scatter and absorb the laser beams, leading to worse results. Please see our supplementary materials for more qualitative results.

## 6 Conclusion

In this work, we propose `RadarOcc`, a novel 3D occupancy prediction approach based on 4DRTs output from 4D imaging radar, enabling robust all-weather perception for autonomous vehicles. We analyse the rationale and challenges of using 4DRTs for 3D occupancy prediction and present tailored solutions to cope with the large, noisy and spherical 4DRTs. Experiments on the K-Radar dataset show `RadarOcc`'s state-of-the-art performance in radar-based 3D occupancy prediction and comparable results to other modalities in normal weathers. Through qualitative analysis, we also exhibit its unique robustness against various adverse weathers. We believe our work could endorse the potential of 4D imaging radar to be an alternative to LiDAR and setup an effective baseline for further research and development of 4D radar-based occupancy perception.

**Limitations.** As an initial investigation into 4D radar-based 3D occupancy prediction, this work has several limitations as follows. First, our method maps single-frame 4D radar data to single-frame 3D occupancy prediction without modeling the temporal information and performing occupancy forecasting. Second, due to the lack of point-wise annotation, our task is limited to two general semantics, *i.e.,* foreground and background. Future work will aim to address these issues.

## Acknowledgement

This research is partially supported by the Engineering and Physical Sciences Research Council (EPSRC) under the Centre for Doctoral Training in Robotics and Autonomous Systems at the Edinburgh Centre of Robotics (EP/S023208/1).

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

## Appendix

The appendix is organized as follows:

- Section A illustrates more details on our experiment setup, including ground truth generation, dataset statistics, evaluation area and computation resources we used for our experiments.
- Section B introduces implementation details of different components in `RadarOcc`.
- Section C gives more experimental results, visualization and failure case of `RadarOcc`.

Besides, please refer to our supplementary video for more qualitative results.

## A Experiment setup details

**Ground truth generation.** Our pipeline of 3D occupancy annotation is similar to those in [3, 25, 26, 105]. First, we split each LiDAR point cloud from a sequence into the background and foreground part with the help of 3D bounding box annotations. For the background, we superimpose all LiDAR points by transforming them into a united world coordinate using their extrinsic. For the foreground part, we track the same instances (indicated by the same tracking IDs) across the sequence and transform LiDAR points association to them into the coordinates of their bounding boxes. In this way, sparse LiDAR point clouds can be significantly densified and the occupancy labels can be more realistic. Note that K-Radar [42] only annotates the objects in the front of the car. To avoid the interference of moving objects in the back, we only use the front part of each LiDAR sweep for ground truth generation. Second, we transform the background and objects point sets into the current frame coordinate system with respect to the ego-pose of the current frame and the objects' pose. Lastly, we concatenate the background and objects points at the current frame and voxelized the merged point cloud to generate the occupancy labels. In cases where voxels are overlaid or boundaries are not clear, we use the majority voting to decide voxel-wise semantics (foreground vs. background).

**Dataset statistics.** In adverse weather conditions (*e.g.,* fog, rain and snow), water droplets or snowflakes can scatter or absorb LiDAR beams, reducing the effective range of LiDAR and inducing noise in the data. To ensure the high fidelity of our occupancy labels, we select 24 sequences collected in decent weather conditions from K-Radar [42] for annotation and leave the rest sequences collected in poor weathers unannotated, which can only be used for qualitative analysis. We split the annotated 24 sequences into the training, validation and test sets with a ratio of 17:2:5, resulting in 11,333, 1,059 and 2,878 frames, respective. Over 0.5 billion voxels are obtained from all annotated frames, among which free, background and foreground class accounts for 92.3%, 7.4% and 0.3% individually.

**Evaluation area.** As claimed in the main paper, we only report the evaluation results within the area where the horizontal FoV (hFov) of all sensors overlap. This scheme can lead to a more fair comparison as it avoids assessing the hallucinated voxels beyond hFoV for modalities like radar and camera, whose hFoVs cannot fully cover our defined RoI volume ahead of the car. Specifically, the overleap hFoV of K-Radar [42] sensor suite is $107°$, symmetrically distributed around the front axis. The ratio between the final evaluation area and our RoI is calculated as: $1 - \cot(107°/2)/4 \approx 0.812$.

**Computation resources.** All of our experiments are conducted on a Ubuntu server equipped with 2 Nvidia RTX 3090 - 24GB GPUs, an Intel i9-10980XE CPU @ 3.00GHz and a 64GB RAM. The training of our method `RadarOcc` uses 17.98GB VRAM, and takes approximately 16.7 hours.

**License for K-Radar.** The K-Radar dataset [42] is published under the CC BY-NC-ND License, and all K-Radar codes [3] are published under the Apache License 2.0.

## B Implementation details of RadarOcc

**Data volume reduction.** The volume size of input raw 4DRT $\mathbf{V}$ is $256 \times 107 \times 37 \times 64$ $(R \times A \times E \times D)$. By encoding the Doppler bins for each spatial location into 8-channel descriptors, we reduce the size of 4DRTs by $\times \frac{D}{8}$, leading to a 3D spatial data volume with a size of $(256 \times 107 \times 37) \times 8$ with the Doppler axis as the feature dimension. For sidelobe-aware spatial sparsifying, we select the top-$N_r$ ($N_r = 250$) elements per range. The resulting lightweight sparse RT $\mathbf{T}$ per frame is $\sim$5MB. Please refer to Sec. C.1 for how we select the optimal $N_r$.

---

[3] https://github.com/kaist-avelab/K-Radar

**Range-wise self-attention.** In our spherical-based feature encoding, the range-wise self-attention is performed on the non-empty elements per range, *i.e.,* $t_i \in \mathbb{R}^{N_r \times (8+2)} (i = 1, 2, \ldots, R)$, where $N_r = 250$. The 8-channel Doppler descriptors are considered as the input features while the azimuth and elevation indices are converted to positional embeddings with lookup tables [100]. Specifically, we use two layers of multi-head attention with the embedding dimension set as 32, number of heads as 4 and dropout probability to be 0.1 The output is re-organized to a sparse tensor with a dimension of $RN_r \times (32 + 3)$, where the range, azimuth and elevation index is stored for each non-empty element.

**3D sparse convolution.** We utilize the `spconv` library [106] to implement the sparse convolution layers for our spherical-based feature encoding. This encoding process has two types of operation: *3D Submanifold Convolution* and *3D Sparse Convolution*. 3D submanifold convolution only convolves the active spatial locations without altering the sparsity pattern and spatial resolution, while 3D sparse convolution performs convolution on all active locations, expanding the sparsify pattern and allows for spatial resolution change. We leverage the submanifold convolution as the first and last layer and apply three sparse convolution layers in-between. We set the stride as 2 for the last two of 3D sparse convolution to reduce the spatial dimension. As a result, we obtain a 3D dense feature volume $\mathbf{F} \in \mathbb{R}^{64 \times 27 \times 10 \times C_f} (C_f = 192)$, where the spatial dimension is decreased by $\times 4$.

**Deformable self-attention.** Given feature volume $\mathbf{F}$, we use 3D deformable self-attention [61] to augment its spatial features. Two attention layers are used and the number of sampling points of each query is set to 8. Each attention layer has 8 heads and apply a dropout of a rate of 0.1 to the output features. The refined feature volume $\mathbf{F}_r$ has the same dimension as $\mathbf{F}$, *i.e.,* $64 \times 27 \times 10 \times C_f$.

**Spherical-to-Cartesian feature aggregation.** To aggregate features extracted in the spherical coordinates, we build learnable voxel queries $\mathbf{H} = \{h^q\}_q$ with a dimension of $14 \times 128 \times 128 \times C_f$ defined in the LiDAR Cartesian coordinates system. By transforming the 3D points $p^q$ corresponding to our voxel queries $h^q$ into the radar spherical coordinates, we construct a list of 3D reference points $\Phi(p^q)$. Then, the deformable cross attention is used to aggregate the spherical features to Cartesian by considering the spherical volume $\mathbf{F}_r$ as the key and value of attention and the voxel queries $\mathbf{H}$ as the query of the attention. Just as the self-attention module, we use the 3D version of the deformable attention [61], with the same network settings. The dimension of the output Cartesian feature $\mathbf{G}$ have the same size as the learnable queries $\mathbf{H}$, which is $14 \times 128 \times 128 \times C_f$.

**3D occupancy decoding.** Given the Cartesian voxel features $\mathbf{G}$, we firstly apply the 3D version of ResNet-18 [102] for decoding, resulting in 4 level of feature maps, with $\frac{1}{2}, \frac{1}{4}, \frac{1}{8}, \frac{1}{16}$ of the voxel spatial shape and $80, 160, 320, 640$ for feature dimension respectively. These multi-level features are then upsampled back to the target spatial space $H \times W \times L$ using 3D FPN [103], leading to the final features $\mathbf{G}_d$ with a dimension of $14 \times 128 \times 128 \times 4C_f$. Lastly, we use an MLP with the hidden dimension of [64,64] to reduce the feature channel and predict the occupancy probabilities which are normalized by a *softmax* layer. The output is denoted as $\tilde{\mathbf{O}} \in \{0, 1\}^{H \times W \times L \times (C+1)}$.

**Training loss.** The overall loss function $\mathcal{L}$ used to train our network can be written as:

$$\mathcal{L} = \mathcal{L}_{CE} + \mathcal{L}_{LS} + \mathcal{L}_{scal}^{geo} + \mathcal{L}_{scal}^{sem} \tag{4}$$

Given the ground truth denoted as $\hat{\mathbf{O}} = \{\hat{o}_i \in \{c_0, c_1, \ldots, c_C\}\}_{i=1}^{N_o} (N_o = H \times W \times L)$ and the output $\tilde{\mathbf{O}}$, the cross-entropy loss $\mathcal{L}_{CE}$ can be calculated as:

$$\mathcal{L}_{CE} = -\sum_{i=1}^{N_o} \sum_{c=c_0}^{c_C} w_c \hat{o}_{i,c} \log(\tilde{o}_{i,c}) \tag{5}$$

where $N_o$ is the number of voxels, $c$ and $i$ indexes classes and voxels, $\tilde{o}_{i,c}$ is the predicted logit for $i$-th voxel on the class $c$. $\hat{o}_{i,c} = 1$ if $\hat{o}_i = c$; else, $\hat{o}_{i,c} = 0$. To balance different classes, we use $w_c$ for each class calculated as the inverse of the class frequency in K-Radar [42]. Please refer to [104] and [23] for more details on the lovasz-softmax loss $\mathcal{L}_{LS}$ and scene-class affinity loss $\mathcal{L}_{scal}^{geo}$ and $\mathcal{L}_{scal}^{sem}$ we used in Eq. 4.

**Training details.** We train RadarOcc with 10 epochs using Adam optimizer with a learning rate of 3e-4. The batch size is 1 for each GPU. We follow [3] to use loss normalization to balance the weight of the 4 different losses, and cosine annealing [107] with $\frac{1}{3}$ warm-up ratio is used at the start of the training.

| $N_r$ | IoU (%) | | | mIoU (%) | | | 🟩 BG IoU (%) | | | 🟥 FG IoU (%) | | | fps |
|---|---|---|---|---|---|---|---|---|---|---|---|---|---|
| | 12.8m | 25.6m | 51.2m | 12.8m | 25.6m | 51.2m | 12.8m | 25.6m | 51.2m | 12.8m | 25.6m | 51.2m | |
| 850 | - | - | - | - | - | - | - | - | - | - | - | - | CUDA OOM |
| 650 | 52.5 | 43.9 | 30.6 | 34.4 | **27.2** | 19.7 | 52.1 | 43.7 | 30.4 | 16.7 | **10.7** | 8.9 | 2.9 |
| 450 | 53.9 | 44.3 | 30.9 | **36.8** | 26.9 | **19.9** | 53.7 | 44.0 | 30.6 | **19.9** | 9.7 | **9.2** | 3.1 |
| 250 | **54.1** | **45.1** | **31.9** | 34.0 | 25.7 | 19.1 | **53.7** | **44.8** | **31.6** | 14.2 | 6.7 | 6.6 | 3.3 |
| 50 | 52.7 | 44.5 | 31.9 | 32.6 | 25.8 | 19.4 | 52.6 | 44.3 | 31.5 | 12.5 | 7.3 | 7.3 | 3.6 |

Table 6: Impact of the number of selected top elements per range (*i.e.*, $N_r$) in our sidelobe-aware spatial sparsifying. The results are reported on the validation set. Best result is shown in **bold**.

| $N_d$ | IoU @ 51.2m (%) | mIoU @ 51.2m (%) |
|---|---|---|
| 1 | 30.9 | 18.7 |
| 2 | 28.8 | **19.4** |
| 3 | **31.9** | 19.1 |
| 4 | 31.1 | 18.9 |
| 5 | 30.1 | 18.8 |

Table 7: Impact of the number of reserved Doppler bins for each spatial location (*i.e.*, $N_d$). The results are reported on the validation set. Best result is shown in **bold**.

## C  Additional experiment results

### C.1  Impact of the number of reserved top elements $N_r$

In Sec. 4.3, we propose a sidelobe-aware spatial sparsification technique that selects the top-$N_r$ elements for each individual range rather than the entire dense radar tensor (RT). There is indeed a trade-off between preserving critical measurements and filtering noise/compressing the radar tensor in this process. Excessive compression/filtering may result in the loss of weak reflections, while insufficient compression/filtering increases computational costs and retains some level of noise.

To identify the optimal balance, we conducted a series of experiments varying the number of selected top elements for each range, *i.e.*, $N_r$, and assessed performance and inference speed on the validation set. The results, presented in Table 6, indicate that RadarOcc achieves the best results in half of all metrics on our validation set when $N_r = 250$. Both higher and lower values of lead to suboptimal results, suggesting that $N_r = 250$ strikes the best balance between retaining critical signals and filtering noise. Additionally, the inference speed at $N_r = 250$ is relatively higher compared to configurations with larger values. Therefore, we select $N_r = 250$ for RadarOcc's evaluation on our testing set.

### C.2  Impact of the number of reserved Doppler bins $N_d$

To investigate the effect of the number of preserved top values (*i.e., $N_d$*) among Doppler bins for each spatial location, we conducted a series of experiments by varying $N_d$. As shown in Table 7, the change in $N_d$ does not significantly impact our results. For both efficiency and performance, we chose $N_d = 3$ for our method based on the validation set performance.

This can be explained by the fact that K-Radar [42] wraps around overflow values in Doppler measurements due to the limited Doppler measurement range. For example, Doppler speeds of 3.0 m/s and 6.0 m/s are measured within the range of -1.92 to 1.92 m/s as 3.0 - 3.84 = -0.84m/s and 6.0 - 3.84×2 = -1.68m/s, respectively. This ambiguity means the information from the Doppler axis only marginally improves our model. Consequently, changing hardly affects our performance. Table 2 in our paper also shows that our baseline without Doppler bin descriptor (w/o DBD), which only uses mean power, reflects this minimal impact. However, we believe our Doppler bin encoding method could bring more improvement with other radar sensors that have a larger measurement range.

### C.3  Impact of range-wise self-attention

To verify the effectiveness of the range-wise self-attention used in our sphercial-based feature encoding (*c.f.* Sec. 4.4), we experiment by removing it from the network and show the results in Tab. 8. It can be seen that RadarOcc improves the performance on most metrics by adding the

|  | Method | IoU (%) | | | mIoU (%) | | | 🟩 BG IoU (%) | | | 🟥 FG IoU (%) | | |
|---|---|---|---|---|---|---|---|---|---|---|---|---|---|
|  |  | 12.8m | 25.6m | 51.2m | 12.8m | 25.6m | 51.2m | 12.8m | 25.6m | 51.2m | 12.8m | 25.6m | 51.2m |
| (a) | Ours | **48.8** | **39.1** | 30.4 | **34.3** | **28.5** | **22.6** | **47.9** | **38.2** | 29.4 | **20.7** | **18.7** | **15.8** |
| (b) | Ours w/o RWA | 48.6 | 39.0 | **30.7** | 32.8 | 27.7 | 22.0 | 47.4 | 38.0 | **29.6** | 18.1 | 16.3 | 14.2 |

Table 8: Ablation studies on range-wise self-attention designs of `RadarOcc`.

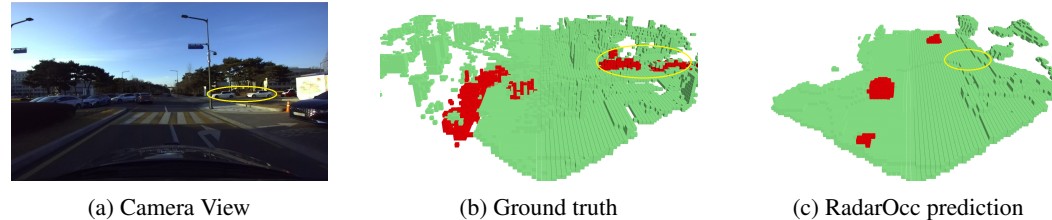

(a) Camera View        (b) Ground truth        (c) RadarOcc prediction

Figure 4: Example of failure case due to insufficient resolution and decreased Signal-to-Noise Ratio at far distances. The white cars parked at the far right are not well predicted.

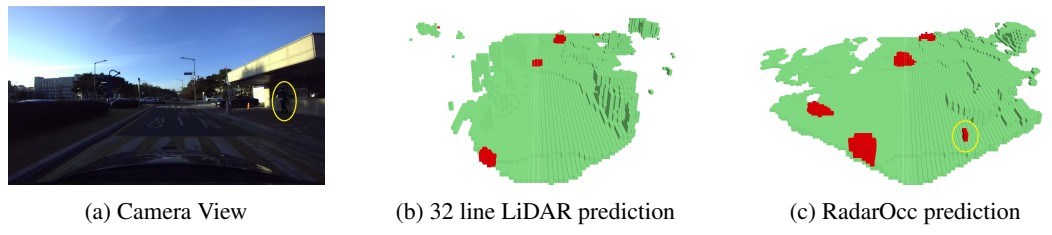

(a) Camera View        (b) 32 line LiDAR prediction        (c) RadarOcc prediction

Figure 5: Example of `RadarOcc` outperforming 32-line LiDAR on objects with low radar cross-section: the pedestrain is recognized.

range-wise self-attention. In particular on FG IoU, the relative gain is 14.4%, 14.7% and 11.3% for the 3D volume of 12.8m, 25.6m and 51.2m, respectively. We credit this to the ability of range-wise self-attention to further suppress the sidelobe noises appearing around the foreground objects.

## C.4 Qualitative results under adverse weather

To better show the qualitative results of `RadarOcc` and baseline methods based on other modalities, we make some video demos under different weather conditions and submit them as a supplementary material. We recommend our audience to watch the video for a better understanding of our work.

## C.5 Example of failure cases

We observed some failure cases of `RadarOcc` due to some reasons, such as insufficient resolution and decreased Signal-to-Noise Ratio (SNR) at far distances. An example of such failure cases is exhibited in Fig. 4. We hope this could shed the light on future research in this field.

## C.6 How we handle object with low radar cross-section

In our method, we address objects with low radar cross-section (RCS) from two key perspectives:

**Input perspective.** We utilize 4D radar tensor (4DRT) data instead of radar point clouds for 3D occupancy prediction. This approach avoids the loss of weak signal returns that can occur during the point cloud generation process, *e.g.*, those filtered out by the CFAR detection, preserving more measurements from low RCS objects compared to radar point clouds.

**Method perspective.** Our sidelobe-aware spatial sparsifying technique selects the top-elements for each individual range rather than the entire dense RT. As shown in Fig. 2, this method retains critical measurements scattered across different ranges, including both low and high RCS objects. This contrasts with percentile-based methods, which often concentrate on elements corresponding to high RCS objects, thereby missing important data from low RCS objects.

As a result, our method is effective in recognizing objects with low RCS, such as pedestrians, when predicting 3D occupancy. Figure 5 shows an example where `RadarOcc` successfully handles low-RCS objects while 32-line LiDAR not.

