# OpenReview forum: "RadarOcc: Robust 3D Occupancy Prediction with 4D Imaging Radar"
_NeurIPS.cc/2024/Conference — NeurIPS 2024 poster_

### Official Review · Reviewer_XRVn · 2024-07-09

**Soundness:** 3
**Presentation:** 4
**Contribution:** 3
**Rating:** 7
**Confidence:** 3

**Summary:**

This paper introduces a novel approach for 3D occupancy prediction in autonomous driving using 4D imaging radar sensors. Traditional methods rely heavily on LiDAR or camera inputs, which are vulnerable to adverse weather conditions. RadarOcc leverages the robustness of 4D radar data, which provides comprehensive scene details even in challenging weather.

The key contributions of the paper include:

- **Utilization of 4D Radar Tensors (4DRT):** Unlike previous methods that use sparse radar point clouds, RadarOcc processes raw 4DRTs, preserving essential scene details and capturing comprehensive environmental data.
- **Novel Data Processing Techniques:** The paper introduces Doppler bins descriptors, sidelobe-aware spatial sparsification, and range-wise self-attention mechanisms to handle the large and noisy 4DRT data. It also uses spherical-based feature encoding followed by spherical-to-Cartesian feature aggregation to minimize interpolation errors.
- **Benchmarking and Evaluation:** RadarOcc is benchmarked on the K-Radar dataset against state-of-the-art methods using various modalities. The results show that RadarOcc outperforms in radar-based 3D occupancy prediction and demonstrates promising results compared to LiDAR and camera-based methods. The approach also shows superior performance in adverse weather conditions.

**Strengths:**

1. The writing is very good, and the demo is provided for the convenience of reviewers to see the effect
2. First work use 4D imaging radar in 3D Occ. The authors propose several novel techniques, such as Doppler bins descriptors, sidelobe-aware spatial sparsification, and range-wise self-attention mechanisms, to efficiently process and utilize the voluminous and noisy 4D radar data. These methods demonstrate creativity in addressing the unique challenges posed by radar data.
3. Extensive experimental and ablation studies.

**Weaknesses:**

1. maybe high computational demand, lack of this part description
2. Dependence on a Single Dataset
3. The experiment was limited because of the data set

**Questions:**

1. In my understanding, the advantage of radar compared with lidar is that it can achieve good perception even in bad weather. However, because of GT procurability,   author just use well-condition sequences, which i think may be unfair to lidar-based methods or vision-based methods, because the domain of radar changes little while others change a lot. It would have been better if the authors had added data on nuscenes to the baseline for comparison. Time may not be enough to complete, but this is really a point of concern to me
2. Could you provide the overlap ratio between different sensors and ROI?
3. Could you provide the overall inference speed and the speed of each module? Additionally, how much does the Doppler bins descriptor reduce the overall computational load, and do the range-wise self-attention and spherical-to-Cartesian feature aggregation introduce significant computational overhead?
4. Since Occ gt is difficult to obtain, is there any way to improve training data and generalization

**I'm willing to raise the grade if address my concerns**

**Limitations:**

As explained by the authors in the limitations section:

1. The model does not fully utilize the 4D radar information.
2. Due to the lack of point-wise annotations in the dataset, the semantic information is limited.

---

> ### Author Response · Authors · 2024-08-07
>
> Dear Reviewer XRVn:
>
> We appreciate your detailed summary and positive comments regarding the presentation, novelty, creativity, and the experimental results of our research. Thanks a lot for providing the valuable feedback and raising insightful questions about our work. We address and answer your questions below:
>
>
> **Q1: Could you provide the overall inference speed and the speed of each module? Additionally, how much does the Doppler bins descriptor reduce the overall computational load, and do the range-wise self-attention and spherical-to-Cartesian feature aggregation introduce significant computational overhead?**
>
> A1: Thank you for raising this concern. Please first refer to our general answer to the **‘Computation complexity and requirement’** question in the global rebuttal.
>
> Specifically, the range-wise self-attention module is highly efficient, with a runtime of only 2.5ms. The spherical-to-Cartesian feature aggregation initially costs 72ms per frame. However, this can be optimized to 29.7ms by reducing the number of layers. This module not only converts the spherical feature volume to a Cartesian one without interpolation errors but also enhances the encoding of voxel feature relationships.
>
> The Doppler bins descriptor significantly reduces the data volume of raw 4D radar tensors (4DRTs) by a factor of D/8, where D equals 64 in our case. This reduction decreases the input data volume to 1/8 of its original size. This reduction not only shortens loading times (from disk/sensor to processors) but also lessens the overall computational burden during spherical-based feature encoding.
>
>
>
> **Q2: Dependence on a Single Dataset. The experiment was limited because of the data set**
>
> A2: Thanks for highlighting this concern. Please refer to our general answer to the **‘Dependence on a single dataset’** answer in the global rebuttal.
>
>
> **Q3: Author just use well-condition sequences, which I think may be unfair to lidar-based methods or vision-based method**
>
> A3: Thanks for this thoughtful feedback. We are not sure if we understand this question exactly. If you were concerned that the training data used to train baselines method does not include sequences collected from adverse weathers, which causes worse performance due to the domain change of LiDAR and camera data, we are happy to answer this question as follows.
>
> Currently, it is hard to resolve this problem as the LiDAR data is inevitably affected by the adverse weather. For example, water droplets or snowflakes can scatter or absorb LiDAR beams, reducing the effective range of LiDAR and inducing noise in the data. As a result, the occupancy labels obtained from accumulated LiDAR data under adverse weather have low fidelity. The nuScenes dataset also faces a similar issue where occupancy labels are affected by adverse weather, despite having annotated bounding boxes. Adding new nuScenes data collected under adverse weather to train our LiDAR and vision-based baselines might not be feasible. This approach could potentially hinder network training because of noisy labels rather than improve adaptation to adverse weather. Moreover, as nuScenes does not provide 4D radar tensor data,  it’s unfair for our radar-based methods because they cannot benefit from the new data samples.
>
> This situation highlights a significant research question: how can we generate reliable 3D occupancy labels under adverse weather conditions when LiDAR measurements are noisy? We acknowledge the importance of this question and plan to address it in our future work.
>
>
> **Q4: Could you provide the overlap ratio between different sensors and ROI?**
>
> A4: Thanks for this comment. Our evaluation focuses on the overlap area between the horizontal field of view (FoV) of all sensors and our defined RoI to minimize potential data discrepancies beyond the FoV. Specifically, the overleap hFoV of K-Radar sensor suite is 107◦, symmetrically distributed around the front axis. The ratio of between the final evaluation area and our RoI can be calculated as:
>
> $$ 1 - \frac{\cot\left(\frac{107^\circ}{2}\right)}{4} \approx 0.812 $$
>
>
> which means 81.2% area of our RoI is taken into account for our final evaluation area.
>
>
>
> **Q5: Any way to improve training data and generalization**
>
> A5: Thanks for this insightful question. It aligns with our research interest in efficiently training 3D occupancy prediction models. Some recent works are exploring self-supervised methods in occupancy prediction. One notable example is EmerNeRF (https://emernerf.github.io/), which can represent highly dynamic scenes in a self-sufficient manner. This approach has the potential to label occupancy ground truth using RGB cameras without requiring human annotation efforts.

---

> > ### Comment · Reviewer_XRVn · 2024-08-13
> >
> > Thank you to the authors for their response. I have carefully reviewed the feedback from the other reviewers as well as the corresponding rebuttal. While the rebuttal has addressed most of my concerns, the limited experimental results prevent me from raising my score. I believe the score I have given is appropriate for this submission.
> >
> > It is a commendable piece of work, and I look forward to seeing more research in this area.

---

### Official Review · Reviewer_f9yn · 2024-07-10

**Soundness:** 3
**Presentation:** 3
**Contribution:** 3
**Rating:** 4
**Confidence:** 4

**Summary:**

This paper leverages recent advancements in automotive radar technology and introduces a novel approach that utilizes 4D imaging radar sensors for 3D occupancy prediction. The proposed method incorporates Doppler bin descriptors, sidelobe-aware spatial sparsification, and range-wise self-attention mechanisms to effectively manage and mitigate the noise present in 4D radar data.

**Strengths:**

This paper presents a compelling algorithm designed to enhance radar-based 3D occupancy prediction by incorporating Doppler information and sidelobe-aware techniques. The authors propose an innovative pipeline that effectively reduces data volume, mitigates sidelobe measurements, and utilizes interpolation-free feature encoding and aggregation.

**Weaknesses:**

Please consult the question section for further information.

**Questions:**

1. Dataset Limitation: The experimental analysis is thorough, but it is conducted solely on the K-Radar dataset. There is a concern that the method may be overfitted to this specific dataset. Could the authors provide additional examples using other datasets to demonstrate the generalizability of the method?

2. Computational Complexity: The methodology includes both an attention mechanism and multi-scale 3D convolutional blocks. A potential issue is whether this structure significantly increases the computational requirements. Could the authors provide a comparison of model size and computational time to address this concern?

3. Selection Criteria: The method selects the Top-3 power values and their corresponding indices. How does this choice affect performance? Would increasing or decreasing the number of selected values significantly impact the results?

4. Failure Cases: Are there any documented failure cases of the method? Considering that Doppler effects are most pronounced when the relative velocity is aligned with the wave direction, how does the method perform in scenarios where this condition is not met?

5. Would the code be published together with the paper?

**Limitations:**

Please consult the question section for further information.

---

> ### Author Rebuttal · Authors · 2024-08-07
>
> Dear reviewer f9yn:
>
> Thank you for acknowledging our method to be compelling and innovative. We are grateful for your insightful questions and feedback. We provided detailed explanations and responses as follows:
>
>
> **Q1: Dataset limitation: The experimental analysis is thorough, but it is conducted solely on the K-Radar dataset. There is a concern that the method may be overfitted to this specific dataset. Could the authors provide additional examples using other datasets to demonstrate the generalizability of the method?**
>
>
> A1:  Thanks for providing this feedback. Please refer to our general answer to the ‘Computation complexity and requirement’ question in the global rebuttal.
>
>
>
> **Q2: Computation complexity: The methodology includes both an attention mechanism and multi-scale 3D convolutional blocks. A potential issue is whether this structure significantly increases the computational requirements. Could the authors provide a comparison of model size and computational time to address this concern?**
>
>
> A2: Thanks for raising this concern. Please refer to our general answer to the ‘Computation complexity and requirement’ question in the global rebuttal.
>
>
> **Q3: Selection criteria: The method selects the Top-3 power values and their corresponding indices. How does this choice affect performance? Would increasing or decreasing the number of selected values significantly impact the results?**
>
> A3: Thank you for this thoughtful question. To investigate the effect of the number of preserved top values ($N_d$) among Doppler bins for each spatial location, we conducted a series of experiments by varying $N_d$. As shown in the table below, the change in $N_d$ does not significantly impact our results. For both efficiency and performance, we chose $N_d$=3 for our method based on the validation set performance.
>
> | $N_d$ | IoU @ 51.2m (%) | mIoU @ 51.2m (%) |
> |-----|-----------------|------------------|
> | 1   | 30.9            | 18.7             |
> | 2   | 28.8            | 19.4             |
> | 3   | 31.9            | 19.1             |
> | 4   | 31.1            | 18.9             |
> | 5   | 30.1            | 18.8             |
>
>
> This can be explained by the fact that K-Radar wraps around overflow values in Doppler
> measurements due to the limited Doppler measurement range. For example, Doppler speeds of 3.0 m/s and 6.0 m/s are measured within the range of -1.92 to 1.92 m/s as 3.0 - 3.84 = -0.84m/s and  6.0 - 3.84*2 = -1.68m/s, respectively. This ambiguity means the information from the Doppler axis only marginally improves our model. Consequently, changing $N_d$ hardly affects our performance. Table 2 in our paper also shows that our baseline without Doppler bin descriptor (w/o DBD), which only uses mean power, reflects this minimal impact. However, we believe our Doppler bin encoding method could bring more improvement with other radar sensors that have a larger measurement range.
>
>
>
> **Q4: Failure cases: Are there any documented failure cases of the method? Considering that Doppler effects are most pronounced when the relative velocity is aligned with the wave direction, how does the method perform in scenarios where this condition is not met?**
>
> A4: Thank you for this comment. We did not observe any failure cases solely due to the loss or weakness of the Doppler effect. The information encoded by the Doppler bins serves as a complement to our primary feature, the mean power values, which reflect the overall measurement intensity at each spatial location. Even without Doppler bins descriptors (reducing the Doppler axis via average-pooling), our method performs well for 3D occupancy prediction, as shown in Table 2. However, we do observe some failure cases of RadarOcc due to other reasons, such as radar sensors suffering from insufficient resolution and decreased Signal-to-Noise Ratio at far distances.  The around ~30 IoU and ~23 mIOU indicate that this is not a perfect model. Please refer to Figure 1 in the **global rebuttal PDF**.
>
>
> **Q5: Would the code be published together with the paper?**
>
> A5: Thanks for reminding us. As claimed in our contribution list, we will release our code public upon acceptance. We also provide our trained models and tools used to preprocess the K-Radar dataset.

---

> ### Author Response · Authors · 2024-08-14
> **Rebuttal follow up**
>
> Dear reviewer f9yn,
> As the discussion period is approaching its end, we kindly invite you to review our detailed rebuttal. We believe it addresses the concerns you raised in your review. If you find our work satisfactory, we would greatly appreciate it if you could consider raising your score to a positive one. We also welcome any further comments you may have.
>
> Thank you again for your time in reviewing our paper.

---

### Official Review · Reviewer_7z3i · 2024-07-12

**Soundness:** 3
**Presentation:** 3
**Contribution:** 3
**Rating:** 5
**Confidence:** 3

**Summary:**

This paper introduces a 3D occupancy prediction method that, unlike previous radar-based approaches, utilizes 4D imaging radar to leverage additional information. To harness the potential of this under-explored 4D data, the paper tackles challenges such as the large size of raw 4D radar data, inherent noise, and discrepancies in coordinate systems. The proposed method results in improved reconstruction accuracy compared with other baselines.

**Strengths:**

The motivation to further exploit the 4D radar data for occupancy estimation is logical. This paper represents pioneering work in addressing this issue.

The paper is well-crafted, featuring clear figures and well-organized content.

The performance under adverse weather conditions is noteworthy, underscoring a critical perception task for the safety of current autonomous vehicles (AVs).

**Weaknesses:**

Comparing the proposed method to only one published baseline and relying on a single dataset raises concerns about the persuasiveness of the findings.

As a learning-based model, the paper misses an evaluation of the generalization of the learning formulation to datasets out of the domain. This oversight is crucial for assessing the model's applicability across various real-world scenarios.

**Questions:**

Can the proposed learning formulation be extended to incorporate radar methods with other modalities, such as images and lidar?

What is the trade-off relationship between compressing the input data (originally 500MB) and its impact on performance? Providing this information could assist readers in balancing efficiency and accuracy.

MISC

Line 110: The line distance is unusual.

**Limitations:**

The experiments are conducted solely on the K-Radar dataset, which is the only autonomous driving dataset containing 4D radar data. This situation highlights a significant issue: the scarcity of 4D data compared to other perception methods (e.g., images, normal radar data), which limits its applicability in broader contexts.

---

> ### Author Rebuttal · Authors · 2024-08-07
>
> Dear Reviewer 7z3i:
>
> We are greatly encouraged that you found this work has logical motivation, represents pioneering work in the area, features clear figures and well-organized content, and underscores safety-critical perception tasks for AVs. We acknowledge the concerns you've highlighted and would like to offer clarifications:
>
> **Q1: Compare to only one published baseline and using single dataset for experiments**
>
> A1: Thank you for highlighting this. There seems to be a misunderstanding regarding the baselines we used for comparison. As RadarOcc is the first method to utilize 4D radar tensor data for 3D occupancy prediction, we compared our method against point-based methods suitable for radar-based comparisons. Specifically, we used three baseline methods from the OpenOccupancy paper: **L-baseline** and **L-CONet** with radar point cloud input, and **L-baseline** with 4DRT-XYZ input. OpenOccupancy represents state-of-the-art point-based work for 3D occupancy prediction. Other available methods such as LMSCNet and JS3C-Net, while influential, are relatively outdated (2020, 2021).
>
> Additionally, to provide a comprehensive evaluation, we included comparisons with LiDAR-based **L-baseline** and single/stereo camera-based **SurroundOcc** methods. These baselines represent state-of-the-art techniques for their respective input modalities, ensuring a persuasive inter-modality comparison.
>
> Regarding the dataset, please refer to our general answer to the **‘Dependence on a single dataset’** answer in the global rebuttal.
>
>
>
> **Q2: Generalization of the learning formulation to datasets out of the domain**
>
> A2:  Thanks for raising this concern. Our RadarOcc is specifically designed for 4D radar tensor input rather than radar point clouds, which means it cannot be directly applied to other existing 4D radar datasets such as VoD and TJ4DRadSet. However, our algorithm can be readily applied if another 4D radar tensor dataset becomes available.
> One potential difference that might affect generalization is the sidelobe levels, which can vary with different radar designs. Various sidelobe suppression techniques, such as the Hamming window, might be applied during the FFT process. To address this, we can incorporate different levels of sidelobe-aware sparsification into our approach to process the data accordingly. This ensures that our method remains robust and adaptable to variations in radar hardware and design.
> By utilizing this flexible approach, we believe that RadarOcc can generalize well to different 4D radar tensor datasets as they become available, thereby extending its applicability across various real-world scenarios.
>
> **Q3: Extend the learning formulation to incorporate radar methods with other modalities**
>
> A3: Thanks for providing this insightful idea. To incorporate radar methods with other modalities, we can fuse the multi-modal information at the feature level by integrating our **voxel feature G** and features extracted from other sensor data, such as images and lidar point clouds. We plan to explore the multi-modal fusion based method in the future.
>
>
> **Q4: Trade-off relationship between compressing the input data and its impact on performance**
>
> A4: Thanks for this thoughtful comment. There is indeed a **trade-off** between preserving critical measurements and filtering noise/compressing the radar tensor in our sidelobe-aware spatial sparsification process. Excessive compression/filtering may result in the loss of weak reflections, while insufficient compression/filtering increases computational costs and retains some level of noise. To identify the optimal balance, we conducted a series of experiments varying the number of selected top elements for each range, i.e., $N_r$, and assessed performance and inference speed on the validation set.
>
> The results, presented in Table 3 of the ‘global’ rebuttal PDF, indicate that RadarOcc achieves the best results in half of all metrics on our validation set when $N_r$ = 250. Both higher and lower values of $N_r$ lead to suboptimal results, suggesting that $N_r$ = 250 strikes the best balance between retaining critical signals and filtering noise. Additionally, the inference speed at $N_r$ = 250 is relatively higher compared to configurations with larger $N_r$ values. Therefore, we select $N_r$ = 250 for RadarOcc’s evaluation on our testing set.
>
>
> **Q5: The applicability in broader contexts of 4D radar data is limited by its scarcity compared to other perception methods**
>
> A5: Thanks for this feedback. As a reminder, although the K-Radar dataset is the only autonomous driving dataset providing available 4DRT data, there are increasing datasets providing 4D radar point clouds (e.g., VoD, TJ4DRadSet, Dual-Radar, NTU4DRadLM). As an emerging sensor, 4D radar is attracting broad attention from the industry and academia. We believe the scarcity of 4D radar dataset will be addressed very soon.
>
> MISC: Line 110: The line distance is unusual.
>
> Thanks for your careful review and for pointing out the issue with the line distance on line 110. We used the ‘\vspace’ command to adjust the line spacing for compactness in this instance. We acknowledge the importance of maintaining consistent formatting and will ensure that such adjustments are removed in future versions of our work.

---

> > ### Comment · Reviewer_7z3i · 2024-08-13
> > **Comment**
> >
> > Thanks for the authors' response to my concerns. I have no further questions and feel I can keep my positive ratings for the paper.

---

### Official Review · Reviewer_v56z · 2024-07-13

**Soundness:** 3
**Presentation:** 2
**Contribution:** 2
**Rating:** 5
**Confidence:** 5

**Summary:**

It is proposed to utilize 4D imaging 8 radar sensors for 3D occupancy prediction by directly processing the 4D radar tensor, thus preserving essential scene details. RadarOcc innovatively addresses the challenges associated with the voluminous and noisy 4D radar data by employing Doppler bins descriptors, sidelobe-aware spatial sparsification, and range-wise self-attention mechanisms. The demonstration of the RadarOcc’s state-of-the-art performance in radar-based 3D occupancy prediction was carried out on K-Radar dataset.

**Strengths:**

It is well-accepted that radar raw data could provide more information for perception tasks in autonomous driving. This submission follows the same direction to utilize 4D imaging radar measurement.

**Weaknesses:**

It is not the first paper to utilize the 4D imaging radar raw measurement for perception. As mentioned by the authors, they focus on semantics, rather than typical road users detection and classification.
For the reference to 4D imaging radar, please consider cite the following paper
S. Sun and Y. D. Zhang, "4D Automotive Radar Sensing for Autonomous Vehicles: A Sparsity-Oriented Approach," in IEEE Journal of Selected Topics in Signal Processing, vol. 15, no. 4, pp. 879-891, June 2021.

For general automotive radar contribution to autonomous driving, please consider cite the following paper
S. Sun, A. P. Petropulu and H. V. Poor, "MIMO Radar for Advanced Driver-Assistance Systems and Autonomous Driving: Advantages and Challenges," in IEEE Signal Processing Magazine, vol. 37, no. 4, pp. 98-117, July 2020.

**Questions:**

It seems to be a trade-off in the sidelobe-aware spatial sparsification process. Is it interesting to know whether important feature information is lost due to this sparsification. For example, targets with weak reflections, such as the information of road curbs might be lost.

**Limitations:**

As mentioned before, the proposed work does not apply to the general road users detection and classification.

---

> ### Author Rebuttal · Authors · 2024-08-07
>
> Dear reviewer v56z:
>
> We appreciate you for the positive feedback regarding the paper’s approach, innovation and experiment results, and agree that radar raw data could provide more information for perception tasks. We understand your concerns and would like to address your points one by one.
>
> **Q1: It is not the first paper to utilize the 4D imaging radar raw measurement for perception. As mentioned by the authors, they focus on semantics, rather than typical road users detection and classification. For the reference to 4D imaging radar, please consider cite the following paper S. Sun and Y. D. Zhang, "4D Automotive Radar Sensing for Autonomous Vehicles: A Sparsity-Oriented Approach," in IEEE Journal of Selected Topics in Signal Processing, vol. 15, no. 4, pp. 879-891, June 2021.**
>
> A1: Thanks for this comment. In this work, we aim to focus on the 3D occupancy prediction task based on 4D radar tensor data. Compared to traditional road user detection and classification, 3D occupancy offers a detailed open-set depiction of scene
> geometry, not limited to specific object classes and shapes. This capability allows it to address a broader range of corner cases than previous object-based perception approaches. The recommended reference is indeed one of the pioneering works that introduces 4D radar sensing to autonomous driving and has inspired our research in some aspects. To ensure rigor, we will cite this paper in Sec. 3.1, where we introduce the 4D imaging radar.
>
>
> **Q2: For general automotive radar contribution to autonomous driving, please consider cite the following paper S. Sun, A. P. Petropulu and H. V. Poor, "MIMO Radar for Advanced Driver-Assistance Systems and Autonomous Driving: Advantages and Challenges," in IEEE Signal Processing Magazine, vol. 37, no. 4, pp. 98-117, July 2020.**
>
> A2: Thanks for sharing this valuable reference, which systematically reviews advancements and challenges of MIMO radar technology in automotive applications, emphasizing its role in enhancing angular resolution for high-resolution imaging radar systems in L4 and L5 autonomous driving. To ensure our audience gains a solid foundational understanding of MIMO radar, we will cite this paper in the second paragraph of the introduction, where we discuss general automotive radar.
>
>
> **Q3: It seems to be a trade-off in the sidelobe-aware spatial sparsification process. Is it interesting to know whether important feature information is lost due to this sparsification. For example, targets with weak reflections, such as the information of road curbs might be lost**.
>
> A3: Thank you for raising this insightful question.
>
> The potential loss of important feature information during the sparsification process is indeed a critical consideration. To address this, our network design incorporates strategies to mitigate these issues. Specifically, our sidelobe-aware spatial sparsification technique selects the top-$N_r$ elements for each individual range rather than the entire dense radar tensor (RT). As demonstrated in **Fig. 2**, this approach retains essential measurements scattered across different ranges, including both strong and weak reflective objects. This is in contrast to percentile-based methods, which often focus on elements corresponding to highly reflective objects, potentially missing crucial data from weak reflective objects.
> There is indeed a trade-off between preserving critical measurements and filtering noise/compressing the radar tensor in our sidelobe-aware spatial sparsification process. Excessive compression/filtering may result in the loss of weak reflections, while insufficient compression/filtering increases computational costs and retains some level of noise. To identify the optimal balance, we conducted a series of experiments varying the number of selected top elements for each range, i.e.,  $N_r$, and assessed performance and inference speed on the validation set.
> The results, presented in **Table 3** of the **‘global’ rebuttal PDF**, indicate that RadarOcc achieves the best results in half of all metrics on our validation set when $N_r $ = 250. Both higher and lower values of  $N_r$ lead to suboptimal results, suggesting that  $N_r $ = 250 strikes the best balance between retaining critical signals and filtering noise. Additionally, the inference speed at $N_r$ = 250 is relatively higher compared to configurations with larger $N_r$ values. Therefore, we select $N_r$ = 250 for RadarOcc’s evaluation on our testing set.

---

> ### Author Response · Authors · 2024-08-14
>
> Dear reviewer v56z,
> As the discussion period is approaching its end, we kindly invite you to review our detailed rebuttal. We believe it addresses the concerns you raised in your review. We also welcome any further comments you may have.
>
> Thank you again for your time in reviewing our paper.

---

### Official Review · Reviewer_kdzm · 2024-07-13

**Soundness:** 3
**Presentation:** 2
**Contribution:** 3
**Rating:** 5
**Confidence:** 4

**Summary:**

The paper presents RadarOcc, a method that enhances 3D occupancy prediction for autonomous vehicles using 4D imaging radar. It directly processes the 4D radar tensor, aiming to overcome the limitations of sparsity and noise associated with conventional radar processing. The methodology introduces techniques such as Doppler bins descriptors and sidelobe-aware spatial sparsification to improve data integrity and scene detail retention. The evaluation is conducted on the K-Radar dataset, with RadarOcc demonstrating superior performance compared to traditional LiDAR and camera-based methods, particularly under adverse weather conditions.

**Strengths:**

The utilization of 4D imaging radar data directly (instead of converting it to sparse point clouds) allows for a more comprehensive environmental perception.  It is also good to consider different weather conditions.

**Weaknesses:**

1. The processing of dense radar tensors can be computationally expensive, potentially limiting real-time application in less powerful onboard systems.

2. The current framework processes single-frame data, which may not fully capture the dynamic nature of driving environments, possibly affecting prediction reliability in highly dynamic scenarios.

3. As this study focuses on solving real problems in autonomous driving, it is important to thoroughly justify why 4D imaging radar has advantages over LiDAR in addressing issues like occupancy prediction. Can methods related to LiDAR really not solve these problems in various environments? In fact, many autonomous vehicles not only refrain from using mm-wave radar but also do not use LiDAR, managing to solve most issues using only vision.

**Questions:**

1. How does the RadarOcc system perform in terms of real-time processing, and what are the computational requirements?

2. Could the method be integrated with temporal data to predict dynamic changes in the environment, and if so, what modifications would be necessary?

3. Are there specific scenarios or environments where RadarOcc's performance might be significantly reduced, such as urban canyons or areas with high radar interference?

4. How does the system handle objects with low radar cross-section, such as pedestrians or animals?

**Limitations:**

1. The method's high computational demand may not scale well to lower-end processors commonly used in commercial vehicles.

2. The lack of temporal modeling could limit the system's predictive capabilities in highly dynamic environments.

3. It remains unclear how well the method generalizes across different radar systems or configurations not represented in the K-Radar dataset.

---

> ### Author Rebuttal · Authors · 2024-08-07
>
> Dear Reviewer kdzm:
>
> We sincerely thank you for providing valuable comments and raising insightful questions about our work. We are glad that you acknowledge that the utilization of 4DRT data allows for a more comprehensive environmental perception, and our work considers different adverse conditions. Here we answer all the questions and hope they can address your concerns.
>
> **Q1: How does the RadarOcc system perform in terms of real-time processing, and what are the computational requirements?**
>
> A1: Thank you for raising this concern. Please refer to our general answer to **‘Computation complexity and requirement’** question in the global rebuttal.
>
>
> **Q2:  Could the method be integrated with temporal data to predict dynamic changes in the environment, and if so, what modifications would be necessary?**
>
> A2:  Thanks for providing this comment. In this work, we only consider the task of 3D occupancy prediction with single-frame 4DRT data. In general, there are two ways to integrate our method with temporal data to improve the prediction reliability.
>
> One approach is to **accumulate multiple frames** captured by the radar sensor and feed them into the network together to obtain per-frame 3D occupancy prediction. To achieve this, we can incorporate a **temporal attention mechanism** after the spherical-to-Catersian feature aggregation module. The temporal self-attention can be employed to explicitly attend to the voxel features of every frame and output **per-frame spatio-temporal voxel features**, which can be input to our 3D occupancy decoding module for sequential-based prediction.
>
> Another way is to utilize the **information from historical frames** to benefit the 3D occupancy prediction for the current frame. To propagate previous latent information to the current frame, we can add a **temporal update module** after the spherical-to-Catersian feature aggregation module, e.g., LSTM and GRU, that treats the voxel features as the hidden state and update it temporally. The **newly updated voxel features** contain temporal information and can produce more reliable prediction for the current frame.
>
> We would consider one of the aforementioned approaches in our future work and discuss this promising direction in our revised paper.
>
>
> **Q3: As this study focuses on solving real problems in autonomous driving, it is important to thoroughly justify why 4D imaging radar has advantages over LiDAR in addressing issues like occupancy prediction. Can methods related to LiDAR really not solve these problems in various environments? In fact, many autonomous vehicles not only refrain from using mm-wave radar but also do not use LiDAR, managing to solve most issues using only vision.**
>
> A3: Thank you for your thoughtful feedback. While LiDAR and cameras perform well under normal conditions, they face significant challenges in extreme weather. For example, in our study (as shown in Figure 3 and our demo videos), rain-induced glare obscured camera lenses, and LiDAR struggled to detect certain objects, occasionally missing ground points entirely. These sensor limitations underscore the difficulties autonomous vehicles encounter without mmWave radar in adverse weather. In dynamic and unpredictable settings, 4D imaging radar provides unique robustness against adverse conditions like fog, rain, and snow, making it a crucial component for enhancing the reliability and safety of autonomous driving. In practice, each sensor modality contributes to different scenarios and plays a vital role in safety-critical autonomous driving, with every piece, including our radar, being essential for long-term mobile autonomy.
>
>
>
> **Q4: Are there specific scenarios or environments where RadarOcc's performance might be significantly reduced, such as urban canyons or areas with high radar interference?**
>
> A4: Thank you for raising this concern. In crowded areas like urban canyons, the portion of **multipath reflections** in the raw radar measurement would increase and, and **inter-object** occlusions become severe. As a result, the performance of RadarOcc might be affected. However, some designs of our network have tried to mitigate these issues. For example, our proposed sidelobe-aware spatial sparsifying could reduce the noise level while reserving important measurements, and our feature encoding and aggregation modules allows for more robust feature extraction. We are committed to refining RadarOcc to further address these challenges in the future, ensuring robustness across various environments.
>
>
> **Q5: How the system handle object with low radar cross-section**
>
> A5: Thanks for your valuable comment. In this work, we address objects with low radar cross-section (RCS) from two key perspectives:
>
> **Input Perspective**: We utilize 4D radar tensor (4DRT) data instead of radar point clouds for 3D occupancy prediction. This approach avoids the loss of weak signal returns that can occur during the point cloud generation process, e.g., those filtered out by the Constant false alarm rate (CFAR) detection, preserving more measurements from low RCS objects compared to radar point clouds.
>
> **Method Perspective**: Our sidelobe-aware spatial sparsifying technique selects the top-$N_r$ elements for each individual range rather than the entire dense RT. As shown in Fig. 2, this method retains critical measurements scattered across different ranges, including both low and high RCS objects. This contrasts with percentile-based methods, which often concentrate on elements corresponding to high RCS objects, thereby missing important data from low RCS objects.
>
> *An example of a detected pedestrian is shown in Figure 2 in the global rebuttal PDF.*
>
> **Q6: It remains unclear how well the method generalizes across different radar systems or configurations not represented in the K-Radar dataset.**
>
> A6: Thank you for highlighting this concern. Please refer to our general answer to the **Dependence on a single dataset** answer in the global rebuttal.

---

> ### Author Response · Authors · 2024-08-14
> **Rebuttal follow up**
>
> Dear reviewer kdzm,
>
> As the discussion period is approaching its end, we kindly invite you to review our detailed rebuttal. We believe it addresses the concerns you raised in your review. We also welcome any further comments you may have.
>
> Thank you again for your time in reviewing our paper.

---

### Author Rebuttal · Authors · 2024-08-07

Dear reviewers and ACs,

We would like to express our sincere gratitude for the careful inspection and constructive feedback from all the reviewers. We are glad to see that all of the reviewers in general hold a positive attitude towards our paper in the pre-rebuttal period. For positive comments, our utilization of 4D radar tensor (4DRT) data is well-motivated (reviewer kdzm, v56z, 7z3i, XRVn), our pipeline of tackling challenges associated with 4DRT is innovative (reviewer v56z, f9yn, XRVn), our experiment is extensive and noteworthy (reviewer kdzm, 7z3i, XRVn), and our paper and demo is well-crafted (reviewer 7z3i, XRVm), we appreciate them and will carry them forward in our future work.

For concerns and questions, we address them one by one in the rebuttal for each individual reviewer. Here, we provide our general answers to two general questions raised by the reviewers, including:

**1. Computation complexity and requirement.**

Answer:
To evaluate the efficiency of RadarOcc, we conducted model inference on a single Nvidia GTX 3090 GPU, achieving an average inference speed of approximately 3.3 fps. Although there is still a gap between the real-time application (10fps), our inference speed has surpassed that of many camera-based methods as reported in the Table 8 of https://arxiv.org/pdf/2303.09551.
Further improvements in inference speed can be achieved by reducing network complexity and applying precision reduction techniques, such as converting model precision from Float32 (FP32) to Float16 (FP16). To validate this, we simplified the feature encoding and aggregation modules by reducing some redudancy layer (e.g. number of layers in deformable attention) for efficiency, and converted the computationally intensive 3D occupancy decoding module from FP32 to FP16. These optimizations resulted in a 126% increase in inference speed, reaching approximately 7.46 fps, with only a minimal impact on performance.
Given the increasing computational power of modern embedded GPUs, such as the Nvidia Orin AGX, which can almost rival desktop GPUs like the Nvidia GTX 2090, we believe this enhanced inference speed demonstrates the potential for real-time application of our method in future vehicle systems, especially if further model quantization is applied. Please refer to Tables 1 and 2 in the global rebuttal PDF for detailed changes in performance and runtime for each module. We will include these results as part of the efficiency evaluation in our paper.


**2. Dependence on a single dataset.**

Answer: We share your concerns regarding dataset usage, but currently, this is the best available option. K-Radar is the only dataset that provides publicly available 4D radar tensor (4DRT) data, which is essential for our RadarOcc method. Our method is specifically designed for 4D radar tensor input, rather than the less informative radar point clouds, making it incompatible with other 4D radar datasets like VoD and TJ4DRadSet. However, if another 4D radar tensor dataset becomes available, our algorithm can be directly applied to it.
While our evaluation is limited to the specific radar configuration of K-Radar, using raw radar data helps mitigate the influence of different point cloud generation algorithms (e.g., CFAR) across various radar systems. This suggests that our method could generalize well to other radar sensors that provide data in the same 4DRT format. In this work, our goal is to demonstrate the potential of 4D radar tensors for this task and to raise awareness within the community and among practitioners about the value of this unique data format.

We warmly welcome any additional questions during the discussion period. Once again, we appreciate everyone's effort in reviewing.

---

### Author Response · Authors · 2024-08-13
**Rebuttal Response Follow-Up**

Dear our reviewers,

I am writing to follow up on the rebuttal we submitted for our paper. We understand the review process can be time-consuming but as the discussion deadline is approaching, we would appreciate any updates or feedback at your earliest convenience so that we could do some last-minute clarification if needed.
We have not yet received any feedback and want to ensure that our reviewers have had the opportunity to read our rebuttal, does our rebuttal answer all your questions?

Best regard,
Submission391 authors

---

### Decision · Program_Chairs · 2024-09-25

**Decision:**

Accept (poster)

**Comment:**

This paper explores the use of 4D imaging radar to predict a 3D occupancy map with applications to autonomous vehicles. The authors point out that this modality could be complementary to vision and lidar based approaches particularly in low-light or adverse weather conditions. The paper was reviewed by a panel of experts who felt that the manuscript did a good job of explaining the method and describing its merits. Experimental validation was limited since there is only a single publicly available dataset that has the requisite 4D radar tensor data. This limits the generality of the inferences that can be drawn from the experiments. Nonetheless the consensus opinion of the reviewers was that the manuscript described an interesting new algorithm and that it would be valuable for people in the field to be aware of what could be achieved with this sensor modality.